# FAST FEW-SHOT GRAPH FLOW PREDICTION

## ABSTRACT

Accurate prediction of traffic flow is crucial for optimizing transportation networks, mitigating congestion, and improving urban planning. However, existing approaches like graph neural networks (GNNs) and traffic simulations face challenges in predicting flow for unseen road networks without historical data. Without abundant training data, GNNs often generalize poorly to new graphs, while simulations can be computationally infeasible for large-scale networks. This paper tackles the problem of few-shot traffic flow prediction in unseen road networks. We propose a novel traffic simulation algorithm that efficiently predicts flow based on node and edge attributes. Through theoretical analysis, we demonstrate our approach closely approximates true flow with asymptotically optimal runtime complexity. Experiments on real-world road networks show our simulation algorithm outperforms GNNs for predicting traffic in unseen cities after training on only three cities. While motivated by traffic prediction in road networks, we expect our contributions to have broader applicability to general graph flow prediction problems across domains.

## 1 INTRODUCTION

Flow is a fundamental property of graphs that characterizes the movement or transfer of entities, such as information, resources, or traffic, across the edges of the graph. In many real-world applications, predicting the flow within a graph is crucial for optimizing resource allocation, identifying bottlenecks, and making informed navigation decisions (Chen et al., 2023). Typically, historical flow data in a graph is used to predict graph flow in the future using graph neural networks (GNNs) or simulations. However, predicting flow in large *unseen* graphs without historical data and with a limited number of training graphs can be significantly more challenging due to weaknesses of the most common flow prediction models: 1) GNNs trained on a small number of training graphs often generalize poorly to unseen graphs because flow patterns can drastically vary across regions, 2) simulations can be computationally infeasible on large graphs.

In this work, we aim to tackle the problem of few-shot flow prediction, with a particular focus on the case of traffic flow prediction in road networks. Traffic flow in a road segment is the number of vehicles crossing over the segment in a unit time interval (whether an hour, a day or a year). Traffic flow prediction is a critical task for urban planning, transportation management, and route optimization, as it enables proactive measures to mitigate congestion and improve the overall efficiency of transportation networks. Predicting flow in unseen cities reduces the dependency on extensive data collection, offering a cost-effective solution for urban planning and traffic management. Besides, accurate flow information can help simulate the potential impact of transportation policies (e.g., congestion pricing and low-emission zones), providing valuable insights for decision-makers. Although our motivation and experiments consider this particular use-case, we emphasize that our approach can be applied to *general* graph flow prediction problems.

Our contributions in this paper are threefold:

- We propose a novel traffic simulation algorithm that efficiently predicts traffic flow based on node-level and edge-level attributes in a graph.

- We provide a theoretical analysis demonstrating that our proposed approach can closely approximate the true flow within the graph and achieves asymptotically optimal runtime complexity, making it computationally efficient and scalable.

- We demonstrate that our traffic simulation algorithm outperforms GNNs in predicting traffic flow in unseen cities using data from only *three* training cities, showcasing the effectiveness of our approach in real-world scenarios.

## 2    RELATED WORK ON GRAPH FLOW PREDICTION

Graph flow prediction requires predicting flow (which could represent fluid volume, vehicle count, monetary transactions etc.) along each edge of a network based on graph features provided at the edge and/or node level. Typically, graph flow prediction is done in a time-series setting, with historical graph flow data available for each edge (Medina-Salgado et al., 2022; Nie et al., 2023; Lv et al., 2020; Wang et al., 2020; Chen et al., 2020; Bao et al., 2023; Tang et al., 2020; Wang et al., 2022). Models used to predict graph flow largely fall under two categories: data-driven approaches and simulation-based approaches.

**Data-driven approaches**    Given graph flow data, flow at unobserved timepoints or locations in the graph can be predicted with a wide variety of machine learning models ranging from nearest-neighbor approaches, linear models and tree ensembles to neural networks (Medina-Salgado et al., 2022; Lartey et al., 2021; Sharma et al., 2001). In road networks, these models use topological and geographical features, socioeconomic features, temporal features, and historical traffic data; they don't exploit the spatial dependency of road network. With the emergence of graph neural networks (GNNs) in recent years, flow prediction has increasingly leveraged GNNs due to networks' inherent graph structure. GNNs (Bruna et al., 2013; Kipf & Welling, 2017) are a type of neural network architecture that uses and maintains the graph structure of its input (Lv et al., 2014; Nie et al., 2023; Chai et al., 2018; Peng et al., 2021; Li et al., 2021; Wang et al., 2021; Li et al., 2022). A key advantage of GNNs are their large representational capacity and ability to leverage arbitrary graph-level features. In principle, GNNs may be used to predict graph flow in unseen graphs; however, in practice, GNNs are susceptible to over-fitting (Zhou et al., 2021), yielding poor performance on unseen graphs particularly when there are only a small number of training graphs. Some recent works have investigated few-shot learning on graphs, but methods to generalize to entirely unseen graphs from a small number of training graphs mostly remain limited to meta-learning settings (in which many different graph generalization problems are provided) (Zhang et al., 2022a).

**Simulation-based approaches**    Simulations are an alternative approach used to predict flow properties in a graph. In road networks, traffic simulations can model graphs at different resolution levels, ranging from modeling individual agents on the graph to modeling continuous flows on edges (Dorokhin et al., 2020; Azlan & Rohani, 2018; Pell et al., 2017; Barceló et al., 2010). Relative to GNNs, traffic simulations are able to model flows at a much finer scale, as well as easily incorporate known physical constraints of flow in a network (such as relationship between traffic density and traffic speed along a road segment) (Fritzsche & Ag, 1994). Regardless of resolution level, traffic simulations typically first model origin-destination pairs for agents on the graph, then compute traffic flow by summing together flows of individual routes from origins to destinations (Van Aerde et al., 1996; Pursula, 1999). However, this approach can be infeasible for large graphs as enumerating over all possible origin-destination pairs is computationally expensive. More recent approaches have incorporated neural networks into traffic simulations, aiming to make them more efficient or performant (Zhang & El Kamel, 2018; Gora & Bardoński, 2017; Zhang et al., 2022b). However, these methods do not provide theoretical guarantees and do not reduce the fundamental computational complexity of enumerating over origin-destination pairs.

## 3    EFFICIENTLY PREDICTING GRAPH FLOW

In this section, we present our flow-simulation based approach to efficiently predict graph flow in unseen graphs; this is valuable in settings where collecting graph labels is expensive, requiring generalization from only a small number of labeled graphs. We prove error bounds on the output of our simulation, and demonstrate that it is computationally efficient, requiring only linear time in the size of the graph to predict flow at each edge.

## 3.1 SETUP AND NOTATION

Suppose we have a directed graph with $m$ edges and $n$ nodes. Assume that each edge has $d$ features associated with it; in a traffic graph, these may correspond to road type (e.g. highway, surface road), number of lanes, road width etc. We also assume that each node has $f$ features; these may correspond to the local population density or the identity of any points of interest (POIs) nearby.

We also suppose that each edge has a *flow* associated with it; in a traffic graph, this may correspond to the average traffic flow on the corresponding road segment. We denote the flow in all edges as $Q \in \mathbb{R}^m$. Our goal is to predict $Q$ from the edge and node features.

## 3.2 FLOW SIMULATION

Now, we present our efficient flow simulation algorithm to predict $Q$. In many natural flow prediction problems, flow is the result of agents taking *trips* on the graph; for instance, in a traffic graph, people take trips between origin-destination pairs, and the flow on a road segment is the total number of trips taken on the segment during a given time period. Thus, we assume that flow $Q_i$ for any edge $i$ can be decomposed by counting all origin-destination pairs traversing through the edge:

$$Q_i = \sum_{O,D} \mathcal{N}(O, D) p(i \in S_{O,D}) \tag{1}$$

where $O$ and $D$ denote origin and destination respectively, $\mathcal{N}(O, D)$ denotes the number of trips between $O$ and $D$, $p$ indicates probability and $S_{O,D}$ is a path from $O$ to $D$.

We make two key assumptions: we assume that the count of the number of trips between $O$ and $D$ can be approximated as: $\mathcal{N}(O, D) \approx \phi(O)^T \psi(D)$ where $\phi(O), \psi(D) \in \mathcal{R}^l$ are feature vectors. With sufficiently many features, we may approximate any $\mathcal{N}$ with arbitrary precision. We also assume that for any two points $O$ and $D$, agents traverse the minimum cost path between the points, denoting the cost of the minimum cost path as $c(O, D) \geq 0$. We assume that the costs obey the triangle inequality:

$$c(A, B) \leq c(A, C) + c(C, B) \tag{2}$$

for all $A, B, C$. By convention, we also assume $c(A, A) = 0$. Note that the cost need not be symmetric: $c(A, B) \neq c(B, A)$ in general.

Finally, we assume the cost can be approximated as: $c(O, D) \approx ||O - D||/R$ where $R$ is a constant and $|| \cdot - \cdot ||$ represents a metric between nodes. We may interpret this as $R$ roughly being an 'average' travel speed across the network; thus, $||O - D||/R$ would represent an 'average' travel time between $O$ and $D$. We may then show the following result:

**Theorem 1.** *Assume for all $O, D$, $|\mathcal{N}(O, D) - \phi(O)^T \psi(D)| \leq \epsilon$ and $|c(O, D) - \frac{||O-D||}{R}| \leq \varepsilon$. Finally, assume that for all $O, D$, the cost of the second lowest cost path between them is greater than the first lowest cost by at least $\Delta$. Then, for edge $i$ from $A$ to $B$, we may approximate $Q_i$ as $\hat{Q}_i$:*

$$\hat{Q}_i = \left( \sum_O e^{\frac{\kappa}{R}(||O-B|| - ||O-A|| - Rc_i)} \phi(O) \right)^T \left( \sum_D e^{\frac{\kappa}{R}(||A-D|| - ||B-D|| - Rc_i)} \psi(D) \right) \tag{3}$$

*for a constant $\kappa > 0$ with approximation error bounded as:*

$$|Q_i - \hat{Q}_i| \leq n^2 \epsilon + (n^2 \epsilon + \sum_j Q_j)(e^{-\kappa\Delta} + e^{4\kappa\varepsilon} - 1) \tag{4}$$

See Appendix A for a proof. For each edge $i$, this approximation computes a weighted average of the feature vectors $\phi(O)$ and $\psi(D)$ over all possibilities of origin and destination; each triple of edge, origin and destination is assigned a weight. For a given origin-destination pair, the weighting of an edge depends on two factors: 1) the cost of the edge $c_i$, with higher cost edges being assigned less weight, 2) the 'progress' an edge makes towards the destination and away from the origin, with more progress being assigned more weight. The resulting weighted vectors associated with the edge correspond to the origin and destination characteristics of *all likely paths* passing through the edge. The final flow is then simply computed as the dot product of the two weighted vectors. Intuitively,

this approach assigns less flow to costlier edges, and more flow to edges in the direction of common routes. It is also biased towards assigning more flow on shorter edges: it is costlier to incorrectly predict a longer edge lies on a shortest path compared to a short edge, so the simulation prefers avoiding flow on long edges.

Observe that the approximation error bound approaches zero as $\epsilon$ and $\kappa\varepsilon$ approach zero and $\kappa\Delta$ approaches infinity. We may set $\kappa$ to minimize the bound at $\kappa = \frac{\log\frac{\Delta}{4\varepsilon}}{4\varepsilon+\Delta}$ when $\Delta > 4\varepsilon$, although in practice the $\kappa$ minimizing the bound may not necessarily be the one minimizing the empirical estimation error.

## 3.3 RUNTIME ANALYSIS

We assume that computing feature vectors $\phi$ and $\psi$ takes constant time. Then, computing $\hat{Q}_i$ following Equation 3 takes $O(n)$ time, where $n$ is the number of nodes in the graph. Computing $\hat{Q}$ for all edges takes $O(mn)$ time where $m$ is the number of edges.

In fact, we may show that the per edge cost of $O(n)$ is asymptotically optimal: for any graph size, there exists a graph such that some edge in the graph requires $O(n)$ time to compute its edge flow. Formally,

**Theorem 2.** *Consider all graphs for which edge flow $Q_i$ for all edges $i$ may be decomposed as:*

$$Q_i = \sum_{O,D} \phi(O)^T \psi(D) p(i \in S_{O,D}) \tag{5}$$

*where $O, D$ are a pair of nodes in the graph, $\phi(O)$ and $\psi(D)$ are functions of $O$ and $D$ respectively, and $S_{O,D}$ denotes the lowest cost path between $O, D$. Then, for any number of nodes $n \geq 4$, there exists a set of graphs and associated $\phi, \psi$ with the same nodes, edges and edge costs such that 1) for some edge $i$, $Q_i$ is different for all graphs, 2) for any $n-3$ nodes $a_1, a_2, ...a_{n-3}$ and evaluations of either $\phi$ or $\psi$ on each node, there exist two distinct graphs with the same measured values. Thus, it takes at least $n-3$ measurements of $\phi$ or $\psi$ to identify a graph from the set.*

See Appendix B for a proof. Intuitively, this theorem states that inferring the edge flow at a particular edge can require examining (nearly) all nodes in the graph because all nodes may contribute to the edge flow. We note that there may be certain graphs for which edge flows can be computed more cheaply; however, in the worst case, the time to compute edge flow is $O(n)$.

---

**Algorithm 1** Compute $\hat{Q}$

---

**Require:** Number of origin nodes $n^O$, number of destination nodes $n^D$, number of edges $m$, origin node features $f^{NO} \in \mathbb{R}^{n^O \times d_N}$, destination node features $f^{ND} \in \mathbb{R}^{n^D \times d_N}$, Edge features $f^E \in \mathbb{R}^{m \times d_E}$, Node encoders $G^O, G^D$, Edge encoder $H$, hyper-parameters $R, \kappa$

    $\hat{Q} = 0 \in \mathbb{R}^m$
    $\Phi = G^O(f^{NO}) \in \mathbb{R}^{n^O \times l}$
    $\Psi = G^D(f^{ND}) \in \mathbb{R}^{n^D \times l}$
    **for** edge $i$ **do**
        $c_i = H(f_i^E) \in \mathbb{R}$
        $v^O = 0 \in \mathbb{R}^{n^O}$
        **for** origin node $j$ **do**
            $v_j^O = e^{\frac{\kappa}{R}(||j-i[1]||-||j-i[0]||-Rc_i)}$ // $i[0], i[1]$ are start and end nodes of segment $i$
        **end for**
        $v^D = 0 \in \mathbb{R}^{n^D}$
        **for** destination node $j$ **do**
            $v_j^D = e^{\frac{\kappa}{R}(||j-i[0]||-||j-i[1]||-Rc_i)}$ // $i[0], i[1]$ are start and end nodes of segment $i$
        **end for**
        $\hat{Q}_i = (v^{OT}\Phi)(\Psi^T v^D)$ // Updated flow estimate
    **end for**
    **return** $\hat{Q}$

---

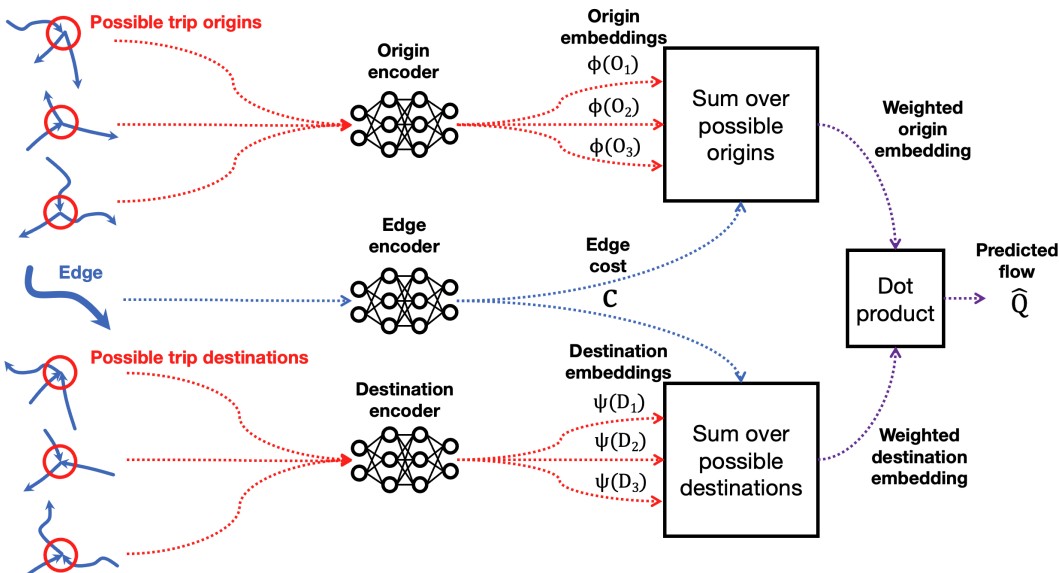

Figure 1: Architecture diagram of our neural network flow simulation. The method works by first encoding edges and roades (representing either origins or destinations) into node embeddings and cost with separate neural network encoders. The resulting origin (or destination) embeddings are then weighted based their cost and spatial location to produce a weighted origin (or destination embedding). The dot product between weighted origin and destination embedding produces the final predicted flow.

### 3.4 NEURAL NETWORK PARAMETERIZATION

So far, we have treated edge costs $c_i$ and feature embeddings $\phi$ and $\psi$ as known. However, in practical problems, we may not know the true costs of traveling on each edge or the likelihood of a trip between each pair of nodes. Instead, we may learn these quantities via neural networks. Specifically, we assume that instead of edge costs and node feature embeddings, we have *properties* for nodes and edges. In a traffic flow graph, these properties may include population density and the popularity of an area for a node, and road width and length for an edge. Given these properties we propose using three neural networks: one to map edge properties to costs and one each to map node properties to either origin or destination node feature embeddings. In our implementation, the two node embedding networks share all but the last layer. Importantly, these neural networks are applied *independently* across edges and nodes respectively (they are not graph neural networks). Given a set of edge flow labels $Q_i$, we may then train these networks to predict flow end-to-end with the simulation algorithm. Figure 1 illustrates the full architecture of the neural network parameterized traffic simulation; observe that the neural networks simply parameterize the inputs to the remaining components of the algorithm. In order to make the algorithm computationally tractable when using neural network predicted feature embeddings, instead of considering all possible origin and destination nodes, we randomly sub-sample possible origin and destination nodes. Algorithm 1 presents the full procedure to compute graph flow from a sample of edge and node features.

## 4 RESULTS

### 4.1 EXPERIMENTAL SETUP

#### 4.1.1 DATASET

We consider a real-world traffic prediction problem: we are provided traffic flow data for three training cities in Texas (Austin, Dallas and San Antonio), and we wish to predict traffic flow in two test cities (Fort Worth and Houston). Traffic flow in road networks is measured as Average Annual Daily Traffic (AADT) (Administration, 2015), the average number of vehicles passing through any given

road segment on each day. Data comes from four sources: 1) OpenStreetMap features, 2) European Space Agency satellite imagery, 3) National Aeronautics and Space Administration (NASA) population density data, 4) road traversal data. The first three sources provide features for the prediction problem, while the last data source provides labels.

**OpenStreetMap**  The OpenStreetMap (OSM) dataset (licensed under the Open Data Commons Open Database License) includes both geographic information about road segments as well as features of each road segment. Road segments are short sections of road often ranging from 10 to 100 meters. OSM lists the geographic coordinates of the points on the road segments as well as their intersections. OSM features include various attributes of the road segments, such as road functional class (e.g., highway, local, residential), presence of stop signs or traffic signals and number of lanes. Most of these features are categorical, with the exception of the following numeric features: speed limit, number of lanes, and the maximum allowed weight and height. Categorical features are one-hot encoded while numerical features are encoded directly. Appendix C Table 4 lists all OSM features. Each city has between $10^5$ and $10^6$ road segments; however, with the exception of road functional class, other features are not available for every segment. For numeric features, we add an additional binary feature indicating whether the numeric feature is available or not.

**European Space Agency satellite imagery**  The European Space Agency provides a number of high-resolution images of different geographic regions (with a Creative Commons licence). Each image is $10980 \times 10980$ and captures a rectangular block (in terms of latitude and longitude coordinates). For each city in our dataset, we collect images overlapping with the city; for all cities, at most two images are sufficient to cover the entire metropolitan area. We then process the images as follows: for each road segment in the dataset, we center the full image to center around the road segment, rotate it such that the road segment points left to right, and then center-crop the image to produce a $24 \times 24$ cropped image. This produces a single cropped image centered at each road segment in the dataset. We then pass this image through a 5-layer convolutional neural network (CNN) to produce a 64-dimensional encoding of the image. This CNN is trained end-to-end with the downstream model.

**Population density data**  The NASA population density data we use provides population density estimates for administrative units, providing a latitude-longitude center point for each administrative unit. To associate these point-wise population densities to road segments, for each point in the population density dataset, we assign its population density to the nearest *road segment intersection*. This provides a population density estimate at each road segment intersection; note that the majority of road segments have no population assigned.

**Road traversals**  We use a third-party road traversal dataset consisting of a large set of traversals of the North American road network. The traversals provide GPS trajectory data with a high-resolution sampling rate of 5 seconds. We apply a Hidden Markov Model map-matching algorithm to align GPS trajectories with corresponding segments onto the OSM road network. Once the raw trajectories are map-matched to OSM segments, we calculate the average daily traversal count for each road segment in each of the Texan cities in our dataset. These volumes serve as our labels. As Figure 2 shows, generally larger road functional classes (such as motorways and primary roads), tend to have larger AADT mean and variance. Moreover, AADT statistics are consistent across different cities. We note that these AADT numbers only represent the subset of the true traffic traveling on the road network since not all vehicles are tracked by our dataset. However, we assume that the sampled vehicles are representative of all vehicles traveling on the road; thus, our AADT label is a reasonable proxy for the 'true' AADT up to a constant scale factor.

### 4.1.2 ARCHITECTURES & BASELINES

We compare our approach with a baseline graph neural network (GNN). Both the GNN and our traffic simulation use a CNN image encoder first to convert satellite images into a lower dimensional representation. This output is then appended with OSM features to form the full set of features for each edge. For GNNs, we append the features associated with the road intersections at the ends of each road segment to the road segment features (thus, each road segment has two sets of road intersection features). Road intersection features consist of population density measurements and

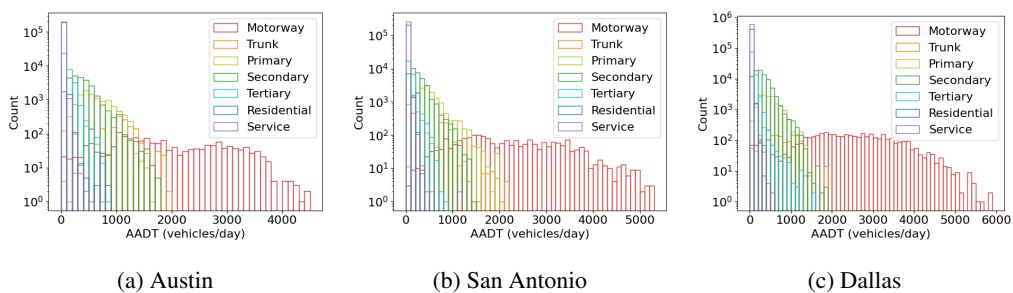

(a) Austin          (b) San Antonio          (c) Dallas

Figure 2: Histograms of AADT for road segments categorized by road functional class in Austin, San Antonio and Dallas.

a positional encoding; these features are used by both the GNN and our method. The GNN is constructed using a graph with *road segments* as nodes and *road intersections* as nodes. This allows the GNN to predict attributes at the road segment level. The GNN uses 3 convolutional layers. For our traffic simulation, the node and edge feature encoders are 8 layer neural networks. See Appendix C for more details.

As an additional baseline, we consider a full traffic simulation that computes flow on each road segment by enumerating over all possible origin and destination node combinations and adding their individual flow contributions. This enumeration is computationally infeasible; thus, we include an estimate of its runtime instead of the actual results with this method. We estimate the runtime by using one floating point operation to account for the flow contributed on each road segment by each origin-destination pair.

### 4.2 Evaluating Prediction Performance

We compare the performance of the GNN vs. our method under a number of error metrics, each split up by road functional class and averaged. Table 1 reveals that the GNN and our method both perform well in an absolute sense, with the RMSE of AADT errors averaging on the order of 100. Comparing to the AADT distribution in Figure 2, these errors can be considered small; they are roughly the size of one histogram bin in the plot.

Nevertheless, our method significantly outperforms the GNN, outperforming the GNN on all error metrics on average and outperforming the GNN on nearly all road functional classes. Appendix D Table 5 shows training set results, where we find that the GNN performs worse than our method on the training set as well. This suggests that the GNN is simply less capable of capturing the structure of traffic flow relative to our traffic simulation which, unlike the GNN, *explicitly models* the structure of traffic flow.

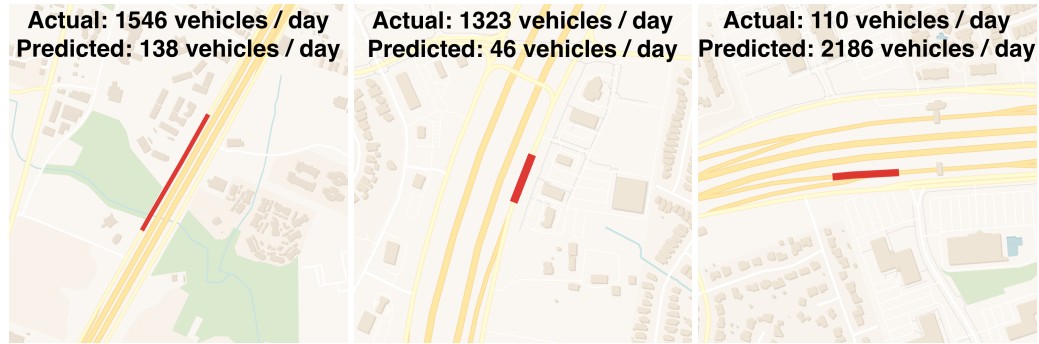

Figure 3: Example road segments in Austin with actual and predicted AADT by our traffic simulation model.

Table 1: Test set AADT error metrics by road functional class and on average for two models trained on Austin, San Antonio and Dallas; root mean squared error (RMSE), mean absolute error (MAE), mean absolute percentage error (MAPE).

| Our Traffic Simulation | | | | | | |
| --- | --- | --- | --- | --- | --- | --- |
| | Houston | | | Fort Worth | | |
| Road Functional Class | RMSE (vehicles/day) | MAE (vehicles/day) | MAPE | RMSE (vehicles/day) | MAE (vehicles/day) | MAPE |
| Motorway | 1067.48 | 816.53 | 1.01 | 1095.47 | 848.83 | 0.39 |
| Trunk | 395.33 | 307.06 | 0.60 | 382.13 | 314.28 | 2.88 |
| Primary | 354.86 | 267.78 | 0.80 | 301.73 | 230.20 | 0.76 |
| Secondary | 203.93 | 148.94 | 0.99 | 196.81 | 140.55 | 0.85 |
| Tertiary | 76.27 | 48.88 | 1.07 | 82.91 | 50.98 | 0.99 |
| Residential | 17.68 | 9.11 | 0.99 | 17.79 | 8.14 | 0.76 |
| Service | 16.52 | 6.54 | 0.77 | 15.60 | 6.07 | 0.62 |
| Average | 102.02 | 28.17 | 0.89 | 91.96 | 24.35 | 0.68 |
| GNN | | | | | | |
| | Houston | | | Fort Worth | | |
| Road Functional Class | RMSE (vehicles/day) | MAE (vehicles/day) | MAPE | RMSE (vehicles/day) | MAE (vehicles/day) | MAPE |
| Motorway | 1911.06 | 1447.67 | 2.31 | 1434.86 | 1138.85 | 0.52 |
| Trunk | 621.43 | 450.15 | 0.85 | 419.16 | 308.37 | 1.40 |
| Primary | 645.37 | 364.53 | 1.09 | 410.79 | 309.95 | 0.66 |
| Secondary | 331.34 | 213.89 | 1.06 | 263.02 | 191.14 | 0.81 |
| Tertiary | 126.98 | 62.45 | 1.33 | 93.01 | 59.06 | 0.88 |
| Residential | 84.74 | 20.10 | 2.64 | 22.54 | 9.40 | 0.85 |
| Service | 68.93 | 12.36 | 1.92 | 22.31 | 7.67 | 0.96 |
| Average | 167.79 | 39.91 | 2.26 | 100.84 | 27.88 | 0.90 |

Figure 3 shows representative road segments where the our traffic simulation either dramatically overestimates or underestimates traffic flow. In all three cases, the road segment connects to a highway on-ramp or off-ramp, with dramatic variation in both the actual and predicted AADTs despite similar road topology. This suggests our approach struggles with AADT prediction on highways which align with our tabulated metrics: errors are largest for major roadways, with RMSEs on the order of 1000 vehicles/day.

## 4.3 EVALUATING RUNTIME

We compare the per epoch training time of our traffic simulation with the GNN and the baseline full traffic simulation (Appendix C describes our computing infrastructure). In Table 2, we find that the GNN trains roughly 10 times slower per epoch than our traffic simulation despite its shallower architecture. This is primarily due to the relatively high cost of graph convolution in the GNN while our approach encodes road segments independently. A full traffic simulation is practically infeasible, requiring on the order of 10 years to run.

Table 2: Projected per epoch training times of three different approaches to compute AADT. Training is conducted on Austin; we estimate per epoch runtimes by projecting runtimes from 10 training iterations. We report mean ± standard deviation over 5 trials.

| | Ours | GNN | Full Traffic Simulation |
| --- | --- | --- | --- |
| Time per epoch (s) | $358.7 \pm 7.7$ | $3348 \pm 35$ | $4.207 \pm 0.135 \times 10^8$ |

## 4.4 ABLATION STUDY

Next, we perform an ablation of our model to to interpret how it works. We test a segment-wise neural network (NN) that replaces the convolutions of a GNN with independent computations for each road segment. This can also be viewed as a version of our traffic simulation that predicts flow purely from edge cost $c_i$ for edge $i$. Table 3 reveals that the segment-wise NN outperforms the GNN, showing that the GNN does not effectively use the connectivity structure of road segments. We also find in Appendix D Table 5 that the segment-wise NN performs closer to our traffic simulation on certain road classes and test cities, sometimes even outperforming it. However, the segment-wise

NN performs much better on the training set than the traffic simulation, indicating that it is prone to overfitting. Thus, we believe our traffic simulation generalizes much better to unseen graphs.

Table 3: Average test set AADT root mean squared error (RMSE) for three models trained on Austin, San Antonio and Dallas and tested on Houston and Fort Worth.

| Method | Our Traffic Sim | GNN | Segment-wise NN |
|---|---|---|---|
| Houston | 102.02 | 167.79 | 120.95 |
| Fort Worth | 91.96 | 100.84 | 79.62 |

## 5 DISCUSSION

We demonstrate theoretically and empirically that our traffic simulation achieves accurate flow predictions under limited run-time. Unlike GNN approaches, which struggle to generalize in few-shot settings, our approach readily predicts flow on unseen graphs *without any additional fine-tuning*. Moreover, it is highly scalable, computing city-scale traffic flows under limited computational resources.

We highlight some potential directions to extend our work. Our current implementation uses a simplified cost approximation between nodes based on their Euclidean distance, which assumes that nodes lie in a Euclidean geometry. Extending this to a neural network parameterized cost function could further improve accuracy. In our work, we also primarily study traffic flows on road networks. We believe that a more flexible cost function could also allow our approach to be applied to general graphs with non-planar and even non-Euclidean geometries.

Given the significant cost of traffic simulation approaches and their widespread use for prediction in transportation, logistics and communication networks, we hope our work can provide an efficient alternative to traditional simulations. More broadly, we hope our approach can facilitate highly efficient and generalize flow prediction on general graphs.

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

## A  PROOF OF THEOREM 1

*Proof.* First, recall that a path between $O$ and $D$ is the lowest cost path if all its segments are on the minimum spanning tree *to* $D$ (denoted $M_{:D}$) and the minimum spanning tree *from* $O$ (denoted $M_{O:}$). Thus, we may write:

$$p(i \in S_{O,D}) = p(i \in M_{:D})p(i \in M_{O:}) \tag{6}$$

where $p$ denotes probability. Note that this factorization is valid for $O \neq D$ since the probabilities are always either 0 or 1. We may then write: $Q_i = \sum_{O,D} \mathcal{N}(O,D)p(i \in M_{:D})p(i \in M_{O:})$ Next, observe that if $i$ connects point $A$ to $B$, then $i$ is on $M_{:D}$ if and only if:

$$c(A,D) = c(B,D) + c_i \tag{7}$$

where $c_i$ is the cost of segment $i$; this is because if $i$ is *not* on the minimum cost path, $c(A,D)$ will be less than the right hand side. Given this, we approximate $p(i \in M_{:D})$ as:

$$p(i \in M_{:D}) \approx e^{\kappa(c(A,D)-c(B,D)-c_i)} \tag{8}$$

for a constant $\kappa > 0$. This has error bounded as:

$$|p(i \in M_{:D}) - e^{\kappa(c(A,D)-c(B,D)-c_i)}| \leq e^{-\kappa\Delta} \tag{9}$$

where equality can be achieved when $i$ is on the second lowest cost path between $A$ and $D$ (when $p(i \in M_{:D}) = 0$). Similarly:

$$|p(i \in M_{O:}) - e^{\kappa(c(O,B)-c(O,A)-c_i)}| \leq e^{-\kappa\Delta} \tag{10}$$

Combining these bounds:

$$|p(i \in S_{O,D}) - e^{\kappa(c(O,B)-c(O,A)-c_i+c(A,D)-c(B,D)-c_i)}| \leq e^{-\kappa\Delta} \tag{11}$$

where equality is reached when each and $p(i \in M_{:D}) = 1$, $p(i \in M_{O:}) = 0$ and the bound on the approximation of $p(i \in M_{O:})$ is tight or vice versa.

Next, we use the approximation on $c(O,D)$:

$$\kappa(c(A,D) - c(B,D) - c_i + c(O,B) - c(O,A) - c_i)$$
$$\approx \frac{\kappa}{R}(||A-D|| - ||B-D|| - Rc_i + ||O-B|| - ||O-A|| - Rc_i) \tag{12}$$

We may bound the approximation error as:

$$|\kappa(c(A,D) - c(B,D) - c_i + c(O,B) - c(O,A) - c_i) -$$
$$\frac{\kappa}{R}(||A-D|| - ||B-D|| - Rc_i + ||O-B|| - ||O-A|| - Rc_i)| \leq 4\kappa\varepsilon \tag{13}$$

Thus,

$$|e^{\kappa(c(A,D)-c(B,D)-c_i+c(O,B)-c(O,A)-c_i)}$$
$$- e^{\frac{\kappa}{R}(||A-D||-||B-D||-Rc_i+||O-B||-||O-A||-Rc_i)}| \leq e^{4\kappa\varepsilon} - 1 \tag{14}$$

with equality when one of the exponents is 0. Combining this bound with the earlier bound on $p(i \in S_{O,D})$:

$$|p(i \in S_{O,D}) - e^{\frac{\kappa}{R}(||A-D||-||B-D||-Rc_i+||O-B||-||O-A||-Rc_i)}| \leq e^{-\kappa\Delta} + e^{4\kappa\varepsilon} - 1 \tag{15}$$

Next, combining this result with the approximation of $\mathcal{N}(O,D)$, we have:

$$|\mathcal{N}(O,D)p(i \in S_{O,D}) - \phi(O)^T\psi(D)e^{\frac{\kappa}{R}(||A-D||-||B-D||-Rc_i+||O-B||-||O-A||-Rc_i)}|$$
$$\leq \epsilon + (\epsilon + \mathcal{N}(O,D))(e^{-\kappa\Delta} + e^{4\kappa\varepsilon} - 1) \tag{16}$$

with equality reached when $p(i \in S_{O,D})$ equals one and the previous inequalities are tight.

Finally, summing over $O, D$ pairs:

$$|\sum_{O,D} \mathcal{N}(O,D)p(i \in S_{O,D}) - \sum_{O,D} \phi(O)^T\psi(D)e^{\frac{\kappa}{R}(||A-D||-||B-D||-Rc_i+||O-B||-||O-A||-Rc_i)}|$$
$$\leq \sum_{O,D} \epsilon + (\epsilon + \mathcal{N}(O,D))(e^{-\kappa\Delta} + e^{4\kappa\varepsilon} - 1) = n^2\epsilon + (n^2\epsilon + \sum_j Q_j)(e^{-\kappa\Delta} + e^{4\kappa\varepsilon} - 1) \tag{17}$$

$\square$

## B    PROOF OF THEOREM 2

*Proof.* Proof by construction: consider a graph with $n$ nodes labeled $W_i$ for $i = 1, ... \lfloor \frac{n}{2} \rfloor - 1$, $Z_j$ for $j = 1, ... \lceil \frac{n}{2} \rceil - 1$ and nodes labeled $Y$ and $Z$. Suppose all $W_i$ have a directed connection to $X$ which is connected to $Y$ which is connected to all $Z$. There are no other edges. Finally, assume that $\phi(W_i)^T \psi(Z_j) \geq 0$ for all $i, j$ and all other $\phi(\cdot)^T \psi(\cdot)$ are 0. Then, the flow $Q_{XY}$ on the edge from $X$ to $Y$ is simply:

$$Q_{XY} = \sum_{i,j} \phi(W_i)^T \psi(Z_j) \tag{18}$$

Now, suppose $\phi(W_i)$ can take value either 0 or value $2^{i-1}$, yielding $2^{\lfloor \frac{n}{2} \rfloor - 1}$ settings of $\phi$. Similarly, suppose $\psi(Z_j)$ can take value either 0 or value $2^{(\lfloor \frac{n}{2} \rfloor - 1)(j-1)}$ yielding $2^{\lceil \frac{n}{2} \rceil - 1}$ possible settings of $\psi$. This yields $2^{(\lfloor \frac{n}{2} \rfloor - 1)(\lceil \frac{n}{2} \rceil - 1)}$ possible settings overall.

Note this allows the $Q_{XY}$ to take any integer value from 1 to $2^{(\lfloor \frac{n}{2} \rfloor - 1)(\lceil \frac{n}{2} \rceil - 1)} - 1$, with every possible setting of $\phi$ and $\psi$ corresponding to a different $Q_{XY}$. Thus, this set of possibilities satisfies condition (1) of the theorem.

Note also that for any $W_i$ and any setting of $\phi$ and $\psi$, we may flip the value of $\phi(W_i)$ from 0 to $2^{i-1}$ or vice versa. Similarly, for any $Z_j$, we may flip the value of $\psi(Z_j)$ from 0 to $2^{(\lfloor \frac{n}{2} \rfloor - 1)(j-1)}$. This yields another valid setting of $\phi$ and $\psi$ with the same values except at either $\phi(W_i)$ or $\psi(Z_j)$.

Observe that for any set of $n - 3$ or fewer nodes, there will be a $W$ or $Z$ node excluded since there are a total of $n - 2$ $W$ and $Z$ nodes. We may choose a $W$ or $Z$ node not in the selected set such that flipping its $\phi$ or $\psi$ embedding will yield another valid setting of $\phi$ and $\psi$. Therefore, condition (2) of the theorem is satisfied.

$\square$

## C    ADDITIONAL EXPERIMENTAL DETAILS

**Image Encoder**    Both the GNN and traffic simulation approaches use a convolutional image encoder to encode images into a 64 dimensional vector. This encoder has 5 ReLU-activated convolutional layers with kernel size 3, stride 2 and hidden dimension 64. This is followed by a final global average pooling layer. The output of the image encoder is appended to the OSM features of each road segment. The image encoder is trained end-to-end with the downstream model.

**Positional encoding of road intersections**    For each road intersection, we produce a 64 dimensional positional encoding as follows: given that the latitude and longitude of a point is $(y, x)$, the positional encoding $\phi \in \mathbb{R}^{64}$ is given as:

$$\phi_i = \sin((\frac{6}{5})^i x), i = 0, 1, ...15 \tag{19}$$

$$\phi_{i+16} = \cos((\frac{6}{5})^i x), i = 0, 1, ...15 \tag{20}$$

$$\phi_{i+32} = \sin((\frac{6}{5})^i y), i = 0, 1, ...15 \tag{21}$$

$$\phi_{i+48} = \cos((\frac{6}{5})^i y), i = 0, 1, ...15 \tag{22}$$

The exponentially spaced range of frequencies $(\frac{6}{5})^i$ allows for encoding of spatial information at varying scales.

**GNN model**    The GNN model consists of three ReLU activated graph convolutional layers of hidden layer size 64. A layer normalization layer preceeds each graph convolution. We perform training over small randomly sampled neighborhoods in the graph; each neighborhood has radius 2 and has 4 neighbors in each step. We use 100 such neighborhoods per training step. The GNN is trained with Adam (Kingma & Ba, 2015) with a learning rate of $10^{-3}$ for 1 epoch. We use the following loss function $(\log(y + 1) - \log(\hat{y} + 1))^2$ where $y$ is the true AADT and $\hat{y}$ is the predicted

AADT. We use a logarithmically scaled AADT in the lose because roads have AADT varying across magnitudes, and higher AADT roads can be reasonably be expected to have higher prediction error as well. We add 1 to avoid taking the logarithm of 0 for any roads with no measured traffic flow.

**Traffic simulation**  We set $\kappa = 1.0$ and $R = 0.01$; these parameters are chosen to fit the training cities. We set the size of the node embeddings to be 128. Because computing flow for all edges at once considering all nodes at once is computationally costly, we take batches of size 10000 over edges and batches of size 1000 over nodes. The edge and node encoder neural networks each have 8 layers and hidden dimension 128. We train the encoder neural networks with Adam with a learning rate of $10^{-4}$ for 5 epochs. We use the same loss function as used to train the GNN model.

**Computing Infrastructure**  We run experiments on a 4.05 GHz CPU with 36 GB of memory.

Table 4: List of OSM features used by models. Categorical features are one hot encoded. RFC indicates road functional class. The last three features (distance, displacement and log curvature) are calculated from the list of latitude-longitude coordinates of each road segment.

| Features |
| --- |
| RFC 0: road |
| RFC 10: motorway |
| RFC 15: motorway_link |
| RFC 20: trunk |
| RFC 25: trunk_link |
| RFC 30: primary |
| RFC 35: primary_link |
| RFC 40: secondary |
| RFC 45: secondary_link |
| RFC 50: tertiary |
| RFC 51: tertiary_link |
| RFC 55: unclassified |
| RFC 60: residential |
| RFC 70: service |
| RFC 71: service, emergency |
| RFC 72: service, drive_thru |
| RFC 75: living_street |
| RFC 85: service, alley |
| RFC 90: unpaved |
| RFC 95: track |
| RFC 100: service, parking |
| RFC 101: service, driveway |
| RFC 102: service, parking_aisle |
| Stop Sign: none |
| Stop Sign: minor |
| Stop Sign: all |
| Traffic Signal: none |
| Traffic Signal: signal |
| Traffic Signal: unknown |
| Traffic Signal: lights |
| Toll |
| Delivery Access Rest. |
| Is Via Segment |
| Max Height |
| No Max Height |
| No Route To |
| Roundabout |
| No Commercial |
| Is Tunnel |
| Restricted Ped. Xing |
| Max Weight |
| No Max Weight |
| Restricted Veh. Xing |
| No Through |
| Lanes in Seg. Dir. |
| No Lanes in Seg. Dir. |
| Is Private Road |
| No Route From |
| Total Lanes |
| No Total Lanes |
| Max Speed |
| No Max Speed |
| Distance |
| Displacement |
| Log curvature |

# D ADDITIONAL RESULTS

Table 5: Training and test set AADT error metrics by road functional class and on average for models trained on Austin, San Antonio and Dallas; root mean squared error (RMSE), mean absolute error (MAE), mean absolute percentage error (MAPE).

**Ours**

| Road Functional Class | Austin (training) | | | San Antonio (training) | | | Dallas (training) | | | Houston | | | Fort Worth | | |
|---|---|---|---|---|---|---|---|---|---|---|---|---|---|---|---|
| | RMSE (vehicles/day) | MAE (vehicles/day) | MAPE | RMSE (vehicles/day) | MAE (vehicles/day) | MAPE | RMSE (vehicles/day) | MAE (vehicles/day) | MAPE | RMSE (vehicles/day) | MAE (vehicles/day) | MAPE | RMSE (vehicles/day) | MAE (vehicles/day) | MAPE |
| Motorway | 892.41 | 670.70 | 1.18 | 820.21 | 626.91 | 0.32 | 836.36 | 632.90 | 0.67 | 1067.48 | 816.53 | 1.01 | 1095.47 | 848.83 | 0.39 |
| Trunk | 523.15 | 420.91 | 0.43 | 318.76 | 241.84 | 0.68 | 344.95 | 271.03 | 0.74 | 395.33 | 307.06 | 0.60 | 382.13 | 314.28 | 2.88 |
| Primary | 252.50 | 193.66 | 0.56 | 257.59 | 181.83 | 0.84 | 234.57 | 175.46 | 0.55 | 354.86 | 267.78 | 0.80 | 301.73 | 230.20 | 0.76 |
| Secondary | 217.91 | 154.00 | 0.66 | 156.38 | 111.80 | 1.39 | 183.19 | 127.99 | 0.86 | 203.93 | 148.94 | 0.99 | 196.81 | 140.55 | 0.85 |
| Tertiary | 83.60 | 52.42 | 0.91 | 82.13 | 52.37 | 0.86 | 64.12 | 37.42 | 0.96 | 76.27 | 48.88 | 1.07 | 82.91 | 50.98 | 0.99 |
| Residential | 20.63 | 8.75 | 0.84 | 18.10 | 8.74 | 0.80 | 13.69 | 6.67 | 0.77 | 17.68 | 9.11 | 0.99 | 17.79 | 8.14 | 0.76 |
| Service | 16.81 | 6.66 | 0.71 | 15.85 | 6.19 | 0.74 | 13.48 | 5.66 | 0.74 | 16.52 | 6.54 | 0.77 | 15.60 | 6.07 | 0.62 |
| Average | 84.21 | 22.71 | 0.74 | 78.53 | 21.61 | 0.77 | 76.01 | 19.87 | 0.74 | 102.02 | 28.17 | 0.89 | 91.96 | 24.35 | 0.68 |

**GNN**

| Road Functional Class | Austin (training) | | | San Antonio (training) | | | Dallas (training) | | | Houston | | | Fort Worth | | |
|---|---|---|---|---|---|---|---|---|---|---|---|---|---|---|---|
| | RMSE (vehicles/day) | MAE (vehicles/day) | MAPE | RMSE (vehicles/day) | MAE (vehicles/day) | MAPE | RMSE (vehicles/day) | MAE (vehicles/day) | MAPE | RMSE (vehicles/day) | MAE (vehicles/day) | MAPE | RMSE (vehicles/day) | MAE (vehicles/day) | MAPE |
| Motorway | 2041.77 | 1311.84 | 1.95 | 1834.14 | 1340.20 | 0.70 | 1872.69 | 1335.05 | 1.11 | 1911.06 | 1447.67 | 2.31 | 1434.86 | 1138.85 | 0.52 |
| Trunk | 648.62 | 518.60 | 0.62 | 383.36 | 254.23 | 0.59 | 437.23 | 325.36 | 1.19 | 621.43 | 450.15 | 0.85 | 419.16 | 308.37 | 1.40 |
| Primary | 445.80 | 289.56 | 0.79 | 350.00 | 225.66 | 0.87 | 317.23 | 213.21 | 0.64 | 645.37 | 364.53 | 1.09 | 410.79 | 309.95 | 0.66 |
| Secondary | 316.89 | 189.02 | 0.79 | 242.88 | 136.25 | 1.06 | 242.48 | 155.18 | 1.03 | 331.34 | 213.89 | 1.06 | 263.02 | 191.14 | 0.81 |
| Tertiary | 130.45 | 59.13 | 1.05 | 109.23 | 56.77 | 1.02 | 89.44 | 43.67 | 1.17 | 126.98 | 62.45 | 1.33 | 93.01 | 59.06 | 0.88 |
| Residential | 36.69 | 10.61 | 0.91 | 29.43 | 10.53 | 0.96 | 23.68 | 8.27 | 0.90 | 84.74 | 20.10 | 2.64 | 22.54 | 9.40 | 0.85 |
| Service | 35.58 | 9.04 | 1.23 | 27.99 | 8.16 | 1.13 | 18.60 | 6.72 | 1.01 | 68.93 | 12.36 | 1.92 | 22.31 | 7.67 | 0.96 |
| Average | 125.83 | 27.35 | 1.02 | 110.34 | 25.14 | 1.03 | 100.77 | 20.95 | 0.98 | 167.79 | 39.91 | 2.26 | 100.84 | 27.88 | 0.90 |

**Segment-wise NN**

| Road Functional Class | Austin (training) | | | San Antonio (training) | | | Dallas (training) | | | Houston | | | Fort Worth | | |
|---|---|---|---|---|---|---|---|---|---|---|---|---|---|---|---|
| | RMSE (vehicles/day) | MAE (vehicles/day) | MAPE | RMSE (vehicles/day) | MAE (vehicles/day) | MAPE | RMSE (vehicles/day) | MAE (vehicles/day) | MAPE | RMSE (vehicles/day) | MAE (vehicles/day) | MAPE | RMSE (vehicles/day) | MAE (vehicles/day) | MAPE |
| Motorway | 695.67 | 560.35 | 1.97 | 973.44 | 791.40 | 0.48 | 889.08 | 697.26 | 1.09 | 1635.17 | 1310.08 | 1.98 | 951.68 | 754.49 | 0.42 |
| Trunk | 296.02 | 220.92 | 0.34 | 196.73 | 128.93 | 0.32 | 233.50 | 171.20 | 0.62 | 493.93 | 392.98 | 0.70 | 324.08 | 239.37 | 1.72 |
| Primary | 177.29 | 130.49 | 0.65 | 210.13 | 145.45 | 0.61 | 186.14 | 140.88 | 0.45 | 486.57 | 375.73 | 0.73 | 310.39 | 233.69 | 0.54 |
| Secondary | 167.40 | 119.75 | 0.56 | 124.20 | 84.26 | 0.68 | 156.95 | 108.55 | 0.65 | 312.04 | 242.44 | 0.93 | 234.91 | 175.39 | 0.82 |
| Tertiary | 68.31 | 44.11 | 0.84 | 70.23 | 44.75 | 0.82 | 57.71 | 33.68 | 0.74 | 92.33 | 63.50 | 0.97 | 84.48 | 54.54 | 0.84 |
| Residential | 19.93 | 8.79 | 0.92 | 17.00 | 8.55 | 0.84 | 13.05 | 6.45 | 0.70 | 19.86 | 11.11 | 0.94 | 17.52 | 8.46 | 0.90 |
| Service | 16.94 | 6.94 | 0.75 | 15.29 | 6.23 | 0.75 | 13.11 | 5.63 | 0.72 | 18.21 | 7.90 | 0.68 | 16.55 | 6.70 | 0.62 |
| Average | 54.62 | 17.47 | 0.77 | 59.21 | 17.10 | 0.74 | 54.47 | 14.90 | 0.68 | 120.95 | 34.60 | 0.79 | 79.62 | 24.13 | 0.73 |

