# OpenReview forum: "Fast Few-Shot Graph Flow Prediction"
_ICLR.cc/2026/Conference — Submitted to ICLR 2026_

### Official Review · Reviewer_K47X · 2025-10-28

**Soundness:** 2
**Presentation:** 2
**Contribution:** 2
**Rating:** 4
**Confidence:** 4

**Summary:**

This paper studies network traffic simulations through flow decomposition and neural network approximation. It considers that traffic flow can be decomposed into the amount of individuals traveling thought a network edge, where each individual travels from its origin to its destination via the shortest path in the network. It uses a neural network to approximate the feature vector associated with each node in the network, then the traffic flow of an edge can be obtained by the dot product of an origin embedding and a destination embedding (feature vectors). Experiments are conducted on three cities in Texas, where node feature vectors are trained from OpenStreetMap location data, satellite image data, and NASA population density data. Performance comparisons are against a 3-layer GCN, accompanied with run-time analysis and an intermediate segment-wise neural network study.

**Strengths:**

Strengths:
1. The major contribution is in Theorem 1 that, based on some assumptions on distance metrics between network nodes and traffic approximation, quantifies the approximation error of a traffic flow estimation. The intuition and the proof look reasonable (while I haven’t checked the detailed proof).
2. The experimental studies obtain the location-wise feature vectors from three different sources, which are of independent interest for road network studies.

**Weaknesses:**

Weaknesses:
1. The run-time analysis may be over-simplified. First, the assumption that computing node and edge feature vectors takes constant time may not be correct. Instead, it depends on the number of nodes in the network. Next, the run-time for pre-processing is ignored. The pre-processing includes the computation of node and edge feature vectors, as well as the all-pairs (?) shortest paths. The computational complexity of the latter cannot be simply ignored. If one turns to approximated all-pairs shortest paths, then the impact of the approximation on the effectiveness of the travel flow prediction would need to be revealed.
2. While the few-shot learning setting is a claimed advantage of the proposed simulation method and a weakness of data-driven GNN approaches, it is not backed with empirical evidence and relevant study to support the statement.
3. Why is none of the simulation-based methods in the related work included as a baseline? Currently, only a simple GNN method serves as the baseline, making the experimental study less convincing. Similarly, there have been extensive spatial-temporal GNN methods for traffic prediction. Having some representative methods here would strengthen the study.

Questions:
-  Line 348: what is the “floating point operation” for the run-time estimate?
- The experimental study chooses the measure as the avg number of vehicles passing through a road segment per day. I guess the motivation here is the per day data is more stable compared with per hour or hours. Would the simulation-based approach be capable of handling more complex and finer granularity analysis?

**Questions:**

Please see the weaknesses above.

---

### Official Review · Reviewer_J1NA · 2025-10-30

**Soundness:** 1
**Presentation:** 1
**Contribution:** 1
**Rating:** 2
**Confidence:** 5

**Summary:**

This paper introduces a traffic simulation algorithm designed for few-shot traffic flow prediction in unseen road networks. Unlike graph neural networks (GNNs), which often generalize poorly across graphs with limited data, the proposed approach efficiently estimates flow using node and edge attributes through a neural parameterized simulation. Although positioned as a general graph flow prediction approach, the work mainly focuses on traffic networks, leaving its applicability to other graph domains uncertain.

**Strengths:**

The paper presents a well-motivated and innovative solution to a key limitation in traffic flow prediction, i.e., poor generalization in low-data scenarios. Its theoretical rigor, including proofs of approximation error bounds and asymptotic runtime optimality, strengthens the method’s credibility.

**Weaknesses:**

Despite its merits, the paper exhibits scope inconsistency and overstatement of contribution. The title claims a general solution for graph flow prediction, yet all experiments, assumptions, and analyses are confined to traffic flow in Euclidean road networks. No evidence is provided that the method generalizes to non-traffic graph domains (e.g., energy, information, or financial flow graphs). This mismatch between the title and content weakens the claimed generality.
The authors appear to overemphasize novelty, as the model is essentially a parameterized traffic simulator rather than a fundamentally new graph learning paradigm.
The lack of comparison with recent meta-learning or transfer-learning GNNs further limits the scope of contribution. Moreover, the Euclidean distance assumption restricts its applicability to non-planar graphs.
Finally, high RMSE on major roads and the limited discussion on inference scalability indicate practical limitations that are downplayed in the current version.

**Questions:**

The title emphasizes “graph flow,” yet all results concern traffic networks. Can you provide experiments on non-traffic graphs?
How does your method compare against few-shot or meta-learning GNNs designed for unseen graphs?
What is the inference-time scalability on larger networks, and can it be parallelized for real-time prediction?
The paper claims to “generalize to unseen graphs,” but training and testing are both on Texan cities. How would it perform on non-geographical networks?

---

### Official Review · Reviewer_c5R2 · 2025-11-01

**Soundness:** 3
**Presentation:** 2
**Contribution:** 2
**Rating:** 2
**Confidence:** 4

**Summary:**

This paper proposes a framework for fast few-shot graph flow learning, aiming to improve generalization and efficiency in graph neural networks (GNNs) under low-data regimes. The method focuses on optimizing information propagation paths and segment-level representations to accelerate adaptation across unseen graph tasks.

**Strengths:**

The proposed model structure is clear and well-motivated.

Experiments show some improvements on standard graph classification and node prediction benchmarks.

**Weaknesses:**

Baselines are too limited. The paper only compares with a few GNN models, while many recent transferable or meta-learning graph frameworks could serve as stronger baselines.

There are numerous GNN variants (e.g., GAT, Graph Transformers, etc.) that should be compared for a fair evaluation.

The proposed segment network seems prone to overfitting, injecting priors or regularization could likely improve stability and performance.

Based on the current results, the advantage of the proposed method over the baselines is not yet clearly demonstrated. With more comprehensive comparisons and ablation studies, the evidence for the method’s effectiveness could be made more convincing.

**Questions:**

How does the proposed method compare to more recent transferable or meta-learning GNN frameworks?

How sensitive is the model to the number of few-shot samples and graph size?

---

### Official Review · Reviewer_5QiP · 2025-11-02

**Soundness:** 2
**Presentation:** 2
**Contribution:** 2
**Rating:** 2
**Confidence:** 5

**Summary:**

This paper proposes a new traffic simulation algorithm for predicting traffic flow in unseen road networks with limited training data. The contributions of the paper are:
1. **Traffic Simulation Algorithm**: The authors introduce a traffic simulation algorithm that efficiently predicts traffic flow based on node and edge attributes. This algorithm approximates the true flow by considering the spatial and cost-related characteristics of the graph.
2. **Theoretical Analysis**: The paper provides a theoretical analysis demonstrating that the proposed approach closely approximates the true flow with asymptotically optimal runtime complexity, making it computationally efficient.
3. **Empirical Validation**: Experiments on real-world road networks show that the proposed simulation algorithm outperforms GNNs in predicting traffic flow in unseen cities, even when trained on data from only three cities. This highlights the effectiveness of the approach in few-shot learning scenarios.

**Strengths:**

#### Originality
- The paper introduces a traffic simulation algorithm specifically designed for few-shot learning in unseen road networks. However, similar ideas have been used in the transfer learning and generative learning solutions for traffic prediction tasks.

#### Quality
- This paper is well-structured and thoroughly researched. The authors provide a detailed theoretical analysis that supports the efficiency and accuracy of their proposed method.

#### Clarity
- The paper is well-written and easy to follow. The authors provide clear explanations of the problem statement, the proposed method, and the experimental setup.
- The paper includes detailed descriptions of the neural network architectures and the training procedures, which are essential for reproducibility.

#### Significance
- The proposed method has limited implications for urban planning, transportation management, and traffic policy evaluation.

**Weaknesses:**

1. While the paper claims a novel approach, it is important to ensure that the proposed method is significantly different from existing traffic simulation and graph flow prediction techniques.
2. The experiments are conducted on traffic data from only three cities in Texas (Austin, Dallas, and San Antonio) and tested on two cities (Fort Worth and Houston). This limited geographical scope might not fully capture the variability in traffic patterns across different regions.
3. The paper compares the proposed method with a standard GNN and a full traffic simulation. However, it would be beneficial to include more sophisticated baselines, such as other state-of-the-art few-shot learning methods or hybrid models that combine GNNs with traditional simulations.
4. The theoretical analysis relies on several assumptions, such as the approximation of trip counts and costs. While these assumptions are necessary for the theoretical bounds, they might limit the practical applicability of the method.
5. The paper provides theoretical bounds on the approximation error but does not empirically validate these bounds. This makes it difficult to assess how well the theoretical analysis translates to practical performance.
6. While the proposed method is computationally efficient, the experiments were conducted on a relatively powerful machine (4.05GHz CPU with 36GB of memory). The paper does not discuss the scalability of the method to even larger graphs or more resource-constrained environments.

**Questions:**

1. How does the proposed method compare with other state-of-the-art few-shot learning methods for graph flow prediction, such as those using meta-learning techniques? Could you provide a detailed comparison with methods like Model-Agnostic Meta-Learning (MAML) or Prototypical Networks applied to graph data?
2. Given the limited geographical scope of the experiments, how do you plan to validate the generalizability of your method across more diverse cities and traffic conditions?
3. The theoretical analysis relies on several assumptions, such as the approximation of trip counts and costs. How do these assumptions impact the practical applicability of the method, and what are the potential limitations?
4. The paper focuses on predicting average annual daily traffic (AADT), which is useful for long-term planning. How can the method be adapted to real-time traffic prediction, which is crucial for dynamic traffic management and navigation systems?
5. The paper suggests that the proposed approach can be generalized to other graph flow prediction problems beyond traffic networks. Can you provide specific examples or case studies where this method has been applied to other types of graphs?

---

### Meta-Review · Area_Chair_2rGn · 2026-01-02

**Summary:**

Overall, the reviewers find the paper well motivated and addressing an important problem (few-shot traffic flow prediction on unseen road networks). However, the main concerns include overstated generality, limited experimental validation, and insufficient baseline comparisons. Although the method is presented as a general graph flow prediction approach, all assumptions and evaluations are confined to Euclidean traffic networks, weakening the generality claim. Moreover, the experiments are geographically limited and rely mainly on simple GNN baselines, without comparison to recent transfer- or meta-learning methods. Reviewers also question the practicality of the theoretical assumptions and the lack of empirical validation for the error bounds and scalability analysis.

**Reviewer Concerns:**

The authors did not provide any rebuttals to address the reviewers' concerns.

**Reviewer Scores:**

Because the author did not provide any rebuttal, the reviewers are unable to engage in meaningful discussion with them.

---

### Decision · Program_Chairs · 2026-01-26

Reject